# How does the digital economy affect the green development of China's industry?

Xiujin Guo[1]☯*, Zhengming Wang[2]☯

1 Jingjiang College, Jiangsu University, Zhenjiang, Jiangsu, China, 2 Business College, Nantong Institute of Technology, Nantong, Jiangsu, China

☯ These authors contributed equally to this work.
* guoxiujin@ujs.edu.cn

## Abstract

Digital economy is an important force to promote industrial green development. The purpose of this paper is to explore the impact of digital economy on China's industrial green development and its mechanism of action from 2011 to 2019, and further examines the spatial spillover effect of digital economy on industrial green development using the spatial Durbin model (SDM). The results show that the digital economy can significantly promote the improvement of IGTFP, and the development of the digital economy in the region can drive the green development of industry in the peripheral regions through the spatial spillover effect. Green technology innovation has a partial mediating role in the process of digital economy affecting IGTFP. Industries in central cities can gain greater green development from digital economy development relative to industries in peripheral cities. The above findings remain valid after a series of robustness tests.

## 1. Introduction

Industrialization has contributed to economic development, but it has also led to increased environmental pollution. According to the "Global Environmental Performance Index Report 2022", China ranks 160th out of 180 participating countries and regions with an environmental performance index of 28.40. "The Second National Pollution Source Census Bulletin" released by China's Ministry of Ecology and Environment in 2020, revealed that by the end of 2017, industrial pollution sources accounted for approximately 70% of the country's total number of pollution sources, amounting to 2,477,400. These data not only show China's low environmental performance but also reflect the urgency of accelerating the green transformation of industry. The focus on sustainable economic development and green initiatives is increasing globally. The European Union has made promoting a green transition one of the core elements of its economic recovery plan. Singapore has proposed that all cars and taxis should be powered by clean energy by the year 2030. South Africa and Morocco have attempted to issue green sovereign bonds to promote green investment. The Chinese government has implemented several policies and regulations in response to the green development of industry in the context of the digital economy. These include the "Industrial Green Development Plan (2016–2020)" (2016), the "Regulations on the Implementation of the Law of the

**Data Availability Statement:** All relevant data are in the paper and its Supporting Information files.

**Funding:** This work was supported by the Jiangsu College Philosophy and Social Science research project (2023SJYB2214). The funders had no role in study design, data collection and analysis,

decision to publish, or preparation of the manuscript. The authors have declared that no competing interests exist. All authors approve the manuscript for publication.

**Competing interests:** The authors declare no conflicts of interest.

People's Republic of China on Taxation for Environmental Protection" (2017), the newly amended Law of the "People's Republic of China on Environmental Protection" (2019), and the "14th Five-Year Plan for the Green Development of Industry" (2021). These measures demonstrate the importance that governments attach to the greening of industry and their determination to achieve it.

According to the United Nations Conference on Trade and Development's 'Digital Economy Report' (2022), global Internet bandwidth increased by 35% in 2020, which is the largest increase since 2013. It is anticipated that monthly worldwide data traffic will increase from 230 EB in 2020 to 780 EB in 2026. Furthermore, "The Research Report on the Development of China's Digital Economy" published by the China Academy of Information and Communication Re-search in 2023, predicts that the scale of China's digital economy will account for 41.5% of the country's GDP in 2022. This proportion is equivalent to the share of the secondary industry in the national economy. The current global economy is experiencing a new era of digital transformation, which is considered an emerging industrial revolution. The development of the digital economy offers new opportunities for countries to achieve early economic recovery and creates future economic hotspots [1]. The digital economy can optimize resource allocation, reduce energy intensity and fully empower industrial transformation and upgrading [2–4]. Digital technology promotes the development of industrial structures towards cleanliness and high added value [5]. Theoretically, the digital economy provides a convenient infrastructure for enterprises to promote green technological innovation to realize the greening of production processes, and ultimately to promote the green transformation of industry as a whole [6]. From a practical perspective, World Economic Forum data show that global carbon emissions would be reduced by up to 15 per cent through the implementation of digital solutions in energy, manufacturing, agriculture and land use, buildings, services, transportation and traffic management. It follows that the digital economy may be a completely new program to promote the greening of industry.

Unfortunately, however, the "theoretical dark box" of the mechanisms by which the digital economy influences the greening of industry and the processes by which it does so has not yet been opened. Existing studies discuss the impact of the digital economy on economic development and the environment respectively. Only a small amount of literature has focused on the relationship between the digital economy and greening development. Based on this, this study takes 278 cities in China as the research object, constructs an indicator system to measure the digital economy and industrial green total factor productivity (IGTFP), explores the relationship between the digital economy and industrial green development and the influencing mechanism, and further discusses its spatial spillover effect and heterogeneity effect.

The marginal contributions of this research are mainly the following. First, the digital economy and industrial green development are included in the same analytical framework, and the direct impact of the digital economy on industrial green development is analyzed from both theoretical and empirical perspectives. Second, identifying the mediating mechanisms of green technological innovation and unveiling the "theoretical dark box" of the digital economy's influence on industrial green development. Third, the spatial spillover effect of digital economy on industrial green development was examined using the dynamic Spatial Durbin Model (SDM), and the heterogeneous effect of digital economy on industrial green development was further investigated based on city location and different city classes.

The rest of this paper is organized as follows. Section 2 provides a brief review of the literature on the digital economy and green development. Section 3 explains the mechanism of action. Section 4 is a presentation of the methods and data. Section 5 is a presentation of the empirical results and an analysis of them. Section 6 further discussion on robustness, endogeneity, and heterogeneity. Section 7 draws conclusions and policy im-plications.

## 2. Literature review

Research closely related to this paper includes two main categories, the first of which is the evaluation and drivers of industrial green development. First, traditional total factor productivity measures, such as the Malmquist index proposed by Charnes et al. [7], only take into account the desired output represented by GDP and do not take into account the non-desired output from environmental pollution. In contrast, Green Total Factor Productivity (GTFP) as an important indicator of green development [8], which simultaneously considers the combined efficiency of economic growth, energy consumption and environmental pollution [9], avoids the overestimation of industrial productivity [10]. Therefore, most scholars use green total factor productivity to measure the level of green development, and this paper uses industrial green total factor production (IGTFP) to measure the level of industrial green development. Secondly, the existing literature has extensively explored the factors influencing the green transformation of industry, which can be mainly categorized into three levels: environmental regulation, technology, and upgrading of industrial structure. In terms of environmental regulation, Shapiro and Walker [11] found that there was a significant increase in manufacturing output in the United States from 1990 to 2008. This increase was accompanied by a 60% decline in air pollution emissions, which was mainly due to increased environ-mental regulation. The impacts of different types of environmental regulations on the efficiency of green development of enterprises are heterogeneous [12], appropriate environmental regulation can promote technological innovation of enterprises, thus improving their competitiveness [13], and strict environmental regulation can prevent "pollution transfer" [14]. On the technological front, technological change is the main driver of green development in China's pollution-intensive industries [8]. In terms of industrial structure upgrading, it has a positive impact on environmental efficiency because the tertiary industry consumes less resources and energy and produces less pollution than the secondary industry [15]. Transformation of industrial structure reduces carbon emissions and thus improves environmental performance [16].

The second category discusses the impact of the digital economy on economic development and the environment, respectively. First, in terms of the impact of the digital economy on economic development, at the macro level, the digital economy facilitates increased productivity, which in turn contributes to economic growth [6, 17] and promotes green and sustainable economic development [18]. The digital economy is conducive to promoting China's inclusive green development and realizing high-quality economic development [19]. At the micro level, digital technology facilitates business trade, and business innovation [20–22]. Digitization has led to increased access to alternative financing for micro, small, and medium enterprises (MSMEs), which has led to a significant increase in operational performance, profitability, and productivity of MSMEs, further contributing to the overall trade growth [20]. Second, in terms of the environmental impact of the digital economy, the digital economy plays an important role in carbon emissions, energy structure and energy consumption [23, 24]. ICT investments can substitute for labour and energy inputs [25]. Most studies consider the digital economy as one of the key measures to achieve carbon neutrality, which, in addition to directly leading to lower carbon emissions, also reduces carbon emissions by optimizing resource allocation, upgrading industrial structure, reducing energy intensity and transforming the energy structure [2–4, 26]. For example, Wang et al. [4] found that a 1% increase in the digital economy index results in a 0.886% decrease in CO2 emissions. Zhang et al. [27] suggested that for every standard deviation increase in digital infrastructure, the carbon intensity of the city decreases by 1.9 per cent. Digitalization has a positive impact on energy [28–30]. Shahbaz et al. [29] discovered that a 1% in-crease in the digital economy would result in a 0.021% increase in the consumption of renewable energy and a 0.106% increase in the production of renewable

energy. Xue et al. [31] suggested that the development of the digital economy has led to the optimization of the energy consumption structure. The digital economy plays a very critical role in energy efficiency in both developing and developed countries [32]. Some scholars also hold a different view, arguing that the relationship between the digital economy and carbon emissions is non-linear [33, 34]. For example, Li and Wang [33] argued that the digital economy has an inverted U-shaped relationship with carbon emissions. Even the digital economy can stimulate growth in electricity consumption, which can hinder efforts to reduce carbon emissions [35]. According to Lange et al. [36], the impact of digitization on energy consumption is greater than the impact of reduced energy consumption. Internet development has a positive impact on the scale of energy consumption and a negative impact on the structure and intensity of energy consumption [37]. Ha et al. [38] suggested that digitization has a negative effect on environmental performance improvement in the short term, but a positive effect in the long term.

Although previous studies have not addressed the relationship between the digital economy and industrial green development, research on the digital economy and green development at the provincial, city and enterprise levels can provide lessons for this paper. For example, Lyu et al. [39] found a U-shaped relationship between the digital economy and the green total factor productivity of cities, in which green technological progress, industrial structure upgrading, and factor market distortion are the main channels of influence. Wu et al. [40] argued that the development of the Internet improves the green total factor energy efficiency at the provincial level, and the degree of resource mismatch, regional innovation capacity, and industrial structure upgrading play an important mediating role. Li [41] found that digital transformation promoted economic performance, but showed an inverted U-shaped relationship with environmental performance by examining survey data from 223 Chinese enterprises.

In summary, the existing literature provides a theoretical basis for this paper, but there are still some shortcomings that need to be further improved. First, the existing literature mainly discusses the relationship between digital economy and economic growth and environmental performance separately, and lacks research on the integrated impact of digital economy and economic growth, energy efficiency and environmental performance. Second, the relationship between the digital economy and green development and its impact mechanism have not been investigated from the industry level. Third, in terms of research methodology, the use of general linear regression models may lead to biased results due to the spatial correlation and spillover effects of digital economy and green development. Therefore, this paper integrates the digital economy and industrial green development into the same analytical framework, identifies the mediating mechanism of green technological innovation, examines the spatial spillover effect of the digital economy on industrial green development by using the Spatial Durbin Model (SDM), and further investigates the heterogeneity effect of the digital economy on industrial green development based on different city levels.

## 3. Research hypotheses

### 3.1 The direct impact of the digital economy on the greening of industry

Digital economy is a new type of economy supported by information and communication technology in the era of network intelligence, and is an economy based on digital technology [34, 42]. Most of the literature measures the digital economy from the perspectives of Internet development and digital financial development [9, 28, 33, 43, 44]. Green development is the harmonisation of economic growth, energy efficiency and environmental performance.

Firstly, in terms of internet development, increased internet access will reduce air pollution levels and the spread of energy efficient internet communication technology devices and the

internet will promote more online shopping, teleconferencing, remote working etc., which will lead to reduced energy consumption and consequent reduction in carbon dioxide emissions [45]. The increase in internet use as well as mobile phone subscriptions mitigates carbon emissions in the BRICS countries and contributes positively to environmental quality [46]. Czernich et al. [47] found that broadband infrastructure has a catalytic effect on economic growth, with each 10 percentage point increase in broadband penetration increasing the annual per capita growth rate by 0.9–1.5 percentage points. In turn, economic growth promotes urban green infrastructure, which helps to drive global green growth. Secondly, in terms of digital financial development, the popularisation of digital inclusive finance is conducive to the alleviation of financing constraints of SMEs, thus promoting the green development of SMEs [48]. Under the influence of industrial digital transformation, industrial enterprises tend to adopt cleaner production technologies, and the environmental performance of manufacturing enterprises is significantly improved [49]. Therefore, the following assumption is made.

Hypothesis 1. The digital economy can contribute to the greening of industry.

## 3.2 Indirect impacts of the digital economy on the greening of industry

Digital elements increase the mobility of innovative elements with their low cost, cleanliness, efficiency, ease of replication and high permeability [50]. Broadband Internet has a positive impact on firms' ability to realise process or product innovations, resulting in a higher probability of product innovation [51]. Telecommunication infrastructure can lead to network spillovers, which have the advantage of breaking down spatial barriers, reducing transaction costs and facilitating business model innovation, thus promoting the diffusion and advancement of green technologies [52]. Digital finance facilitates green technology innovation by alleviating the financing constraints of enterprises, promoting industrial structure upgrading, and facilitating the development of the manufacturing industry [53, 54]. Digital technology can help enterprises to research and identify green innovation directions, green innovation potentials and green innovation paths [55]. The digital economy favours green technological progress in the long term [33]. The digital economy can either directly promote green innovation or indirectly promote green innovation by increasing economic openness, optimising industrial structure and expanding market potential [43]. Based on this, the digital economy is favourable to green technological innovation.

Green technological innovation improves the environmental performance of Italian regions [56]. Green innovation contributes to the growth of green total factor productivity of listed firms in China [57]. Green technological innovation capacity facilitates the increase in the utilisation of energy and natural resources as well as the improvement of eco-economic efficiency [58]. Green innovation has a significant contribution to total factor productivity by increasing unit labour productivity [59]. Based on this, green technology innovation positively affects industrial green development.

Digital finance affects corporate environmental performance through green innovation [60, 61]. The information technology revolution represented by the Internet will promote the green development of industry through technological innovation and industrial structure upgrading, and the mediating effect of technological innovation is obviously larger [62]. Green innovation mediates the process by which the digital economy influences green development [39]. This leads to the following hypothesis.

Hypothesis 2. The digital economy indirectly contributes to the greening of industry through the acceleration of green technology innovation.

### 3.3 Spatial spillover effects of the digital economy on the greening of industry

There are extreme inequalities in various digital indicators between developed and developing economies, also known as the "digital divide" [63]. Yilmaz et al. [64] using a panel of 48 U.S. states from 1970 to 1997, found that telecom infrastructure not only has a significant impact on output growth in its own state, but also has spillover effects on other states. Shahnazi and Shabani [65] analysed the impact of communication technology on per capita carbon emissions in Iranian provinces over the period 2001 to 2015 and found that communication technology has a significant spatial spillover effect. Myovella et al. [66] analysed the determinants of the digital divide in Sub-Saharan Africa, concluding that spatial interdependence in Internet access and broadband subscriptions is evident among the 41 geographically interconnected districts in the region. The digital economy is characterised by openness, inter-temporality and economic sharing, which weakens the attenuation law of technological spillovers caused by geographic distances between market participants, and enhances the inclusiveness of information and knowledge, thus generating spillovers [3]. Digital technology not only reduces local carbon emissions, but also reduces carbon emissions from peripheral cities through spatial spillover effects [67]. Therefore, the following hypothesis is proposed.

Hypothesis 3. The digital economy can drive the greening of industries in neigh-boring regions through spatial spillover effects.

## 4. Methodology and data analysis

### 4.1 Econometric methodology

**4.1.1 Basic model.** To study the impact of the digital economy on industrial greening, the following benchmarking model is set as Eq (1):

$$lnIGTFP_{it} = \alpha_0 + \alpha_1 lnDIG_{it} + \alpha_2 Control_{it} + \lambda_i + \theta_t + \varepsilon_{it} \tag{1}$$

where, **$lnIGTFP_{it}$** represents the logarithm of city i's level of industrial greening in period t, **$DIG_{it}$** represents the level of the digital economy of the city i in period t, **Control** represents all control variables, **$\lambda_i$** represents individual fixed effects, **$\theta_t$** represents time-fixed effect, **$\varepsilon_{it}$** is a random error term.

**4.1.2 Intermediary effect model.** To further study the indirect impact mechanism of the digital economy on industrial green development, green technology innovation, and industrial structure upgrading were selected as mediating variables. Referring to the classical mediated effects model proposed by Baron and Kenny (1986) [68], the following mediated effects model was constructed:

$$lnIGTFP_{it} = \alpha_0 + \alpha_1 lnDIG_{it} + \alpha_2 Control_{it} + \lambda_i + \theta_t + \varepsilon_{it} \tag{2}$$

$$lnIGTFP_{it} = \alpha_0 + \alpha_1 lnDIG_{it} + \alpha_2 Control_{it} + \lambda_i + \theta_t + \varepsilon_{it} \tag{3}$$

$$Med_{it} = \beta_0 + \beta_1 lnDIG_{it} + \beta_2 Control_{it} + \lambda_i + \theta_t + \varepsilon_{it} \tag{4}$$

$$lnIGTFP_{it} = \gamma_0 + \gamma_1 lnDIG_{it} + \gamma_2 Med_{it} + \gamma_3 Control_{it} + \lambda_i + \theta_t + \varepsilon_{it} \tag{5}$$

**4.1.3 Spatial correlation test.** Spatial econometric models can only be used when there is a spatial correlation, so Moran's I was utilized to examine whether there is spatial dependence

between the digital economy and industrial green total factor productivity before using spatial econometric models.

$$Moran's\ I = \sum_{i=1}^{n} \sum_{j=1}^{n} W_{ij}(x_i - \bar{x})(x_j - \bar{x}) / (S^2 \sum_{i=1}^{n} \sum_{j=1}^{n} W_{ij}) \tag{6}$$

where, $x_i$ is the observed value of city i, $\bar{x}$ and $S^2$ are its mean and variance respectively, $W_{ij}$ is the spatial weight matrix, which is a reflection of the spatial association characteristics with the geographical inverse distance weight matrix. $-1 < Moran's\ I < 1$. The closer the value is to zero, the smaller the spatial correlation is, $i > 0$, the space is positively correlated; $i < 0$, the space is negatively correlated.

$$W_{ij} = \begin{cases} 1/d, i \neq j \\ 0, i = j \end{cases} \tag{7}$$

In this paper, using the geographic distance matrix, Where d is the geographical distance between city i and city j, calculated based on the latitude and longitude of the city.

**4.1.4 Spatial econometric model.** To further examine the spatial spillover effect of the digital economy on industrial green development, the following spatial Durbin model is constructed (SDM).

$$lnIGTFP_{it} = \alpha_0 + \rho W * lnIGTFP_{it} + \alpha_1 lnDIG_{it} + \alpha_2 Control_{it} + \eta_1 W * lnDIG_{it} + \eta_2 W * Control_{it} \\ + \lambda_i + \theta_t + \varepsilon_{it} \tag{8}$$

Where $\rho$ is the spatial autoregressive coefficient, which indicates the influence of the local industrial green development level on the industrial green development level of neighboring regions, and W is the spatial weight matrix.

## 4.2 Variable selection

**4.2.1 Industrial green total factor productivity.** To avoid the problems of traditional DEA models that do not take into account non-desired outputs and the existence of differences between decision-making units [57]. The Global Malmquist-Luenberger (GML) index [69] based on the SBM model of non-expected output [70] is used to measure IGTFP. The calculation steps according to Yu [71] and Ma and Zhu (2022) [72] are as follows.

(1) Set of global production possibilities.

Assuming that city i is a decision unit ($DMU_i$), and $DMU_i$ contains N kinds of input factors $x = (x_1, \ldots x_n) \in R_N^+$, we can obtain M kinds of expected outputs $y = (y_1, \ldots y_m) \in R_M^+$, and K types of unexpected outputs $b = (b_1, \ldots b_k) \in R_K^+$, so, the global production possibility set is constructed as Eq (9)

$$P^t(x) = \begin{cases} (y^t, b^t) : \sum_{t=1}^{T} \sum_{i=1}^{I} \beta_i^t y_{im}^t \geq y_{im}^t, \forall m; \sum_{t=1}^{T} \sum_{i=1}^{I} \beta_i^t b_{ik}^t = b_{ik}^t, \forall k \\ \sum_{t=1}^{T} \sum_{i=1}^{I} \beta_i^t x_{in}^t \leq x_{in}^t, \forall n; \sum_{t=1}^{T} \sum_{i=1}^{I} \beta_i^t = 1, \beta_i^t \geq 0, \forall i \end{cases} \tag{9}$$

Where, $\beta_i^t$ is the weight of each cross-section.

(2) Global SBM directional distance function.

$$\overrightarrow{s}_v^G = (x^{t,i}, y^{t,i}, b^{t,i}, g^x, g^y, g^b) = max \frac{\frac{1}{N}\sum_{n=1}^{N}\frac{s_n^x}{g_n^x} + \frac{1}{M+K}\left(\sum_{m=1}^{M}\frac{s_m^y}{g_m^y} + \sum_{k=1}^{K}\frac{s_k^b}{g_k^b}\right)}{2} \tag{10}$$

$$s.t. \sum_{t=1}^{T} \sum_{i=1}^{I} \beta_i^t x_{in}^t + s_n^x = x_{in}^t, \forall n;$$

$$\sum_{t=1}^{T} \sum_{i=1}^{I} \beta_i^t y_{im}^t - s_m^y = y_{im}^t, \forall m;$$

$$\sum_{t=1}^{T} \sum_{i=1}^{I} \beta_i^t b_{ik}^t + s_k^b = b_{ik}^t, \forall k;$$

$$\sum_{i=1}^{I} \beta_i^t = 1, \beta_i^t \geq 0, \forall i; S_n^x \geq 0, \forall n; S_m^y \geq 0, \forall m; S_k^b \geq 0, \forall k$$

Where, $(g^x, g^y, g^b)$ denote the directional vectors of input reductions, desired output increases, and undesired output reductions, respectively. $(s_n^x, s_m^y, s_k^b)$ are relaxation vectors that represent the quantities of input redundancy, desired output shortfalls, and undesired output excesses, respectively; if the value is greater than 0, it means that the actual inputs and undesired outputs are greater than the boundary's inputs and outputs, and the desired outputs are less than the boundary's outputs.

(3) GML index and IGTFP.

Based on the directional distance function (DDF), the industrial green total factor productivity (IGTFP) is measured by the Global Malmquist-Luenberger Productivity Index (GML) proposed by Oh [69]. At the same time the index can be decomposed into a technical efficiency change index (GEC), and a technical progress change index (GTC), where GEC refers to improvements in management systems, resource allocation, etc., and GTC refers to improvements in production processes and manufacturing skills, as shown in the following equation.

$$GML_t^{t+1} = [1 + \overrightarrow{s}_v^G(x^t, y^t, b^t; g)]/[1 + \overrightarrow{s}_v^G(x^{t+1}, y^{t+1}, b^{t+1}; g)] = GEC^{t+1} * GTC^{t+1} \quad (11)$$

The GML index represents the change in period t+1 relative to period t. If it is greater than 1, there is an increase in GTFP; if it is less than 1, there is a decrease in GTFP; if it is equal to 1, GTFP is in a steady state. The GML index reflects the growth of GTFP from t to t+1, not GTFP itself, so the base period GML index is set to 1, and the GML is multiplied by the GML of each year in turn to obtain the industrial green total factor productivity for 2011–2019, and the GEC and GTC are calculated according to the same method.

Further, an indicator system is constructed, which consists of inputs, desired outputs, and non-desired outputs, as shown in Table 1. The industrial capital stock is estimated using the

**Table 1. Input-output indicators for IGTFP.**

| Indicator type | Indicator name | Indicator description/unit |
|---|---|---|
| Input | Labor | Total number of employed persons in the secondary industry (10000 people) |
| | Capital | Industrial investment in fixed assets (100 million yuan) |
| | Energy | Industrial electricity consumption (100 million kWh) |
| Expected Output | Industrial output | Gross Domestic Product in the secondary industry (100 million yuan) |
| Unexpected Output | Waster water | Industrial wastewater emissions (10000 tons) |
| | Waster gas | Industrial SO2 emissions (10000 tons) |
| | Solid wastes | Industrial smoke and dust emissions (10000 tons) |

**Table 2. The evaluation index system of the digital economy.**

| Indicator type | Indicator name | Indicator description |
| --- | --- | --- |
| Internet development | Internet penetration rate | Internet users per 100 population |
| | Number of Internet-related employees | Percentage of employees in computer services and software |
| | Number of mobile Internet subscribers | Cell phone subscribers per 100 population |
| | Internet-related outputs | Total telecommunication services per capita |
| Digital financial development | Digital Finance for Inclusive Development | China Digital Inclusive Finance Index |

perpetual inventory method, drawing on Zhang et al. [73], and is measured using 2003 as the base year and a depreciation rate of assets equal to 9.6%.

**4.2.2 Digital economy.** According to Zhao et al. [74] and Huang et al. [75], the digital economy is measured from two perspectives: internet development and digital financial development. Among them, Internet development includes four sub-indicators, namely, Internet penetration rate, number of Internet-related employees, number of mobile Internet users, and Internet-related output. Digital financial development is measured by the China Digital Inclusive Finance Index, jointly compiled by Peking University's Digital Finance Research Center and Ant Gold Service Group. Through the method of principal component analysis, the data of the above five indicators were standardized and downgraded to obtain the comprehensive development index of the digital economy, denoted as DIG. As shown in Table 2.

**4.2.3 Other variables.** To elucidate the indirect impact mechanism of the digital economy on industrial green development, green technology innovation (GTEC) was used as a mediating variable. Green technology innovation is categorised as green product innovation or process-related green process innovation, including energy saving, pollution prevention, waste reuse, green product design, or technological innovation in corporate environmental management [76]. In this paper, we refer to the related research [77], and use the number of green patent applications in cities to indicate the level of green technological innovation in cities.

In order to more accurately assess the comprehensive impact of digital economy on industrial green development and reduce the estimation error caused by omitted variables, referring to related studies [33, 39, 62, 72], this paper selected five control variables, namely, government intervention (GOV), urban economic development (EDL), population size (PSIZE), the level of openness to the outside world (OPEN), and the level of human capital (HCAP) are the five control variables. The above variables are defined in Table 3.

**4.2.4 Data source.** With the introduction of China's "12th Five-Year Plan" in 2011, mobile Internet and mobile e-commerce penetration has increased, and mobile payment has entered into a period of rapid development, which is also a period of rapid development of digitization, so the observation data of 278 cities in China from 2011 to 2019 are selected as the

**Table 3. Definition and expression of variables.**

| Symbols | Variables | Indicator description |
| --- | --- | --- |
| GTEC | Green technology innovation | Urban green patent applications (thousands) |
| GOV | Government intervention | Government fiscal expenditure as a percentage of GDP |
| EDL | Urban economic development | Logarithm of urban GDP per capita |
| PSIZE | Size of the population | The logarithm of the average annual population |
| OPEN | Openness to the outside world level | Total imports and exports as a share of GDP |
| HCAP | Human capital level | The logarithm of the number of students enrolled in university per 10,000 population |

**Table 4. Statistical description of variables.**

| Variable | Definition | Obs | Mean | S.D. | Min | Max |
|---|---|---|---|---|---|---|
| IGTFP | Industrial green total factor productivity | 2502 | 0.9838 | 0.1332 | 0.5432 | 1.6438 |
| DIG | Digital economy | 2502 | 0.6360 | 0.0396 | 0.6000 | 1.0000 |
| GTEC | Green technology innovation | 2502 | 0.5720 | 1.6400 | 0.0010 | 24.4720 |
| GOV | Government intervention | 2502 | 0.1972 | 0.0971 | 0.0439 | 0.9155 |
| EDL | Urban economic development | 2502 | 10.7041 | 0.5694 | 8.8416 | 13.0557 |
| PSIZE | Size of the population | 2502 | 5.8968 | 0.6854 | 3.0057 | 8.1345 |
| OPEN | Openness to the outside world level | 2502 | 0.1859 | 0.3881 | 0 | 9.8664 |
| HCAP | Human capital level | 2502 | 4.7823 | 1.0357 | 0.6931 | 7.1785 |

samples of the empirical study. Green patent data from the State Intellectual Property Office of the People's Republic of China (SIPO). The Digital Financial Inclusion Index originates from the Peking University Institute for Digital Finance. City latitude and longitude data were collected manually through Google Maps to reduce data discretization. City-level data are mainly from the "China Urban Statistical Yearbook", "China Internet Development Statistical Bulletin", "China Energy Statistical Yearbook", provincial and municipal statistical yearbooks, statistical bulletins, National Bureau of Statistics, and Peking University Enterprise Big Data Research Center. For some of the missing data, linear interpolation was used to supplement the data. To summarize, a statistical description of all the variables involved is given in Table 4.

## 5. Results

### 5.1 Direct results

Using the panel data of 278 cities from 2011–2019, we finally chose the FE model to explain the regression results based on the Hausman test. The baseline regression analysis was conducted using stepwise regression and the results are shown in Table 5. The R2 value from column (1) to column (6) is increasing, which indicates that the model fit is increasing with the gradual addition of control variables, indicating that the selection of control variables is reasonable. The regression coefficients of the digital economy on IGTFP are all positive and significant at a 5% level of significance, indicating that the digital economy has a significant positive impact on IGTFP. Hypothesis 1 was supported.

In addition, the results of column (6), which contains all the control variables, show that government intervention (GOV) has a significant negative impact on industrial green development, demonstrating that inappropriate government intervention can hinder industrial green development. Urban economic development (EDL) has a significant positive impact on the green development of industry, and as the level of economic development increases, the public's awareness of green development increases, which in turn promotes the green development of industry. Population size (PSIZE) has a significant positive impact on industrial green development. Cities with a high population density generally also have a higher level of economic development, the public's awareness of green development is stronger, and the city's management of industrial green development is stricter, so the larger the population size, the more it promotes industrial green development. The level of human capital (HCAP) has a significant positive impact on high-quality green development, and the higher the level of human capital, the more favorable it is to the green innovation of industry, thus promoting the green development of industry. Openness to the outside world (OPEN) has a positive impact on industrial green development, but the result is not significant, which may be because the level of openness to the outside world takes a long period to have an impact on industrial green development.

**Table 5. Basic regression results.**

| Variables | ln IGTFP | ln IGTFP | ln IGTFP | ln IGTFP | ln IGTFP | ln IGTFP |
|---|---|---|---|---|---|---|
| | (1) | (2) | (3) | (4) | (5) | (6) |
| ln DIG | 0.639** | 0.566** | 0.559** | 0.676*** | 0.682*** | 0.653*** |
| | (2.37) | (2.13) | (2.16) | (3.02) | (2.98) | (2.85) |
| GOV | | -0.490*** | -0.345*** | -0.296*** | -0.297*** | -0.273** |
| | | (-4.39) | (-3.10) | (-2.82) | (-2.82) | (-2.56) |
| EDL | | | 0.068** | 0.070*** | 0.070*** | 0.076*** |
| | | | (2.59) | (2.73) | (2.71) | (2.87) |
| PSIZE | | | | 0.170** | 0.173** | 0.178*** |
| | | | | (2.55) | (2.56) | (2.64) |
| OPEN | | | | | 0.015 | 0.014 |
| | | | | | (1.16) | (1.15) |
| HCAP | | | | | | 0.010*** |
| | | | | | | (2.61) |
| Constant | 0.292** | 0.342*** | -0.393 | -1.370*** | -1.388*** | -1.551*** |
| | (2.38) | (2.81) | (-1.33) | (-3.15) | (-3.16) | (-3.50) |
| city FE | YES | YES | YES | YES | YES | YES |
| Year FE | YES | YES | YES | YES | YES | YES |
| Observations | 2502 | 2502 | 2502 | 2502 | 2502 | 2502 |
| R-squared | 0.057 | 0.083 | 0.092 | 0.101 | 0.102 | 0.104 |
| F-statistic | 10.69 | 12.28 | 12.67 | 12.87 | 11.90 | 11.44 |

Note: Robust standard errors in parentheses

*** $p < 0.01$

** $p < 0.05$

* $p < 0.1$.

## 5.2 Intermediary effects

Columns (1)-(3) in Table 6 present the results of the mediating effects. Column (1) shows the regression of the independent variable on the dependent variable, which has a coefficient of 0.653 ($p < 0.01$). Column (2) shows that the impact of the digital economy on green technology innovation is significantly positive at 15.850, indicating that the digital economy effectively promotes green technology innovation. In column (3), the regression coefficient of the mediating variable GTEC is 0.020 ($p < 0.01$), and the coefficient of the effect of the digital economy decreases from 0.653 to 0.337 ($p < 0.05$), so it is judged to be a partial mediating effect. In addition, we can conclude that the total effect of the digital economy on high-quality green development is 0.653, and the direct effect is 0.337. the indirect effect is 0.317 ($15.850 \times 0.020 = 0.317$). These findings suggest that the digital economy can enhance industrial greening by facilitating green technological innovation and thereby enhancing industrial greening. Hypothesis 2 was supported.

In addition, a bootstrap test is used, as shown in S1 Table. The confidence interval corresponding to the indirect effect does not include zero, indicating that the mediating effect is significant. This validates that the digital economy can drive industrial green development through green technology innovation.

## 5.3 Spatial effects

**5.3.1 Spatial correlation test.** The geographic distance matrix was used to estimate the spatial results and the results of the spatial correlation test are given in Table 7. Moran's I for

**Table 6. Mediating effects and heterogeneous analysis.**

| Variables | ln IGTFP | GTEC | ln IGTFP | ln IGTFP Central | ln IGTFP Peripheral |
|---|---|---|---|---|---|
| | (1) | (2) | (3) | (4) | (5) |
| ln DIG | 0.653*** | 15.850* | 0.337** | 0.743** | -0.076 |
| | (2.85) | (1.79) | (2.06) | (2.57) | (-0.23) |
| GTEC | | | 0.020*** | | |
| | | | (5.14) | | |
| Constant | -1.551*** | -16.234 | -1.226*** | -0.775 | -1.193** |
| | (-3.50) | (-1.38) | (-3.04) | (-1.05) | (-2.58) |
| Control variables | YES | YES | YES | YES | YES |
| city FE | YES | YES | YES | YES | YES |
| Year FE | YES | YES | YES | YES | YES |
| Observations | 2502 | 2502 | 2502 | 1251 | 1251 |
| R-squared | 0.104 | 0.249 | 0.130 | 0.055 | 0.175 |
| F-statistic | 11.44 | 7.44 | 14.63 | 3.626 | 11.79 |

Note: Robust standard errors in parentheses

*** $p < 0.01$

** $p < 0.05$

* $p < 0.1$.

the digital economy is positive and significant while Moran's I for industrial green development is negative and significant. This indicates that there is a significant spatial correlation between the digital economy and the level of green development of industry, and there is a diffusion effect of the development of the digital economy in the local city on the neighboring cities, while the improvement of the level of green development of the industry in the local city will produce a clustering effect, attracting the industry with a high level of green development in the neighboring cities, which is not conducive to the green development of the industry in the neighboring cities. The Moran'I scatterplot in Fig 1 more intuitively describes the spatial

**Table 7. Results of global Moran's I test.**

| Year | ln DIG | | ln IGTFP | |
|---|---|---|---|---|
| | Moran's I | Z-statistic | Moran's I | Z-statistic |
| 2011 | 0.049* | 1.388 | -0.027* | 1.287 |
| 2012 | 0.049* | 1.366 | -0.057* | -1.324 |
| 2013 | 0.041** | 1.157 | -0.010 | -0.156 |
| 2014 | 0.048* | 1.370 | -0.055* | -1.275 |
| 2015 | 0.042** | 1.647 | -0.103*** | -2.451 |
| 2016 | 0.059* | 1.593 | -0.049 | -1.105 |
| 2017 | 0.047* | 1.286 | -0.105*** | -2.488 |
| 2018 | 0.046* | 1.420 | -0.107*** | -2.526 |
| 2019 | 0.046* | 1.423 | -0.067** | -1.557 |

Note: Robust standard errors in parentheses

*** $p < 0.01$

** $p < 0.05$

* $p < 0.1$.

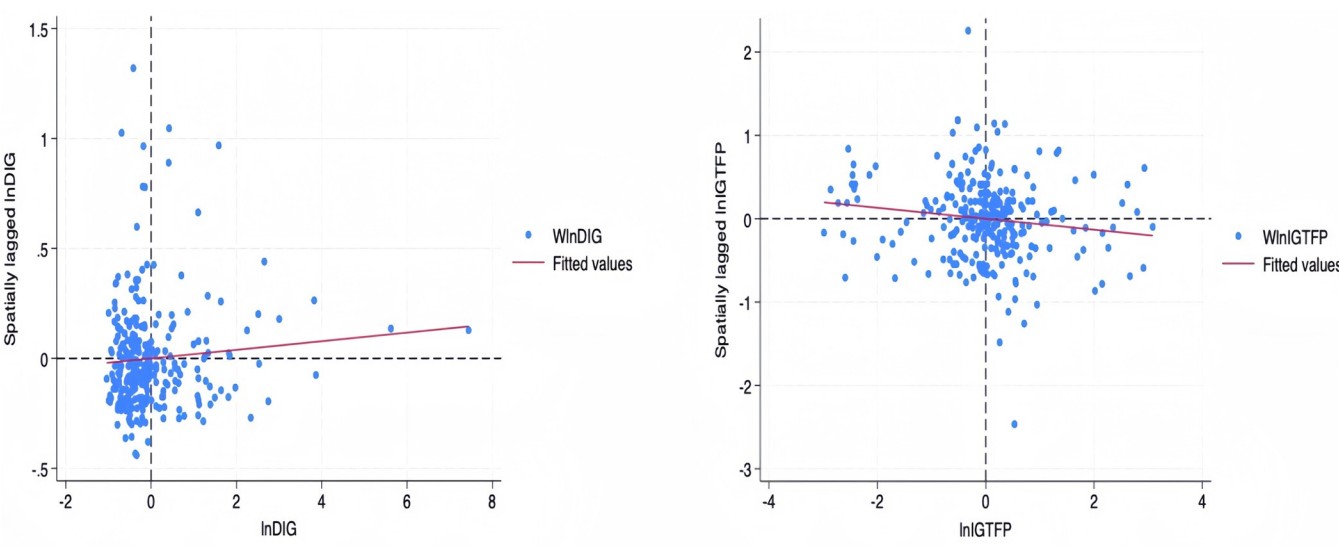

**Fig 1. Moran's I scatter plot for lnDIG and lnIGTFP in 2019.**

characteristics of the digital economy lnDIG and the industrial green development level lnIGTFP in 2019, and it can be observed that the digital economy exhibits significant positive spatial correlation and clustering phenomenon, while the industrial green development level exhibits significant negative spatial correlation and clustering phenomenon.

**5.3.2 Model selection.** According to LeSage and Pace [78], this paper first tested for spatial correlation based on OLS regression through LM and Robust LM tests. The results of the test rejected the hypothesis of spatial independence, which implies that the analysis can be further considered using spatial econometric models [79]. The results of the LR test rejected the hypothesis that the original time or spatial effects were not significant, and finally chose to use the spatio-temporal fixed effect. The results of the Wald test and the LR test showed that the SDM could not be degraded into the SEM or the SLM. All the results are listed in Table 8.

**Table 8. Test results of model selection.**

| Test | Statistic | P-value |
| --- | --- | --- |
| LM(LAG) | 15.087*** | 0.000 |
| R-LM(LAG) | 0.169*** | 0.000 |
| LM(ERR) | 20.835*** | 0.000 |
| R-LM(ERR) | 5.916** | 0.015 |
| Wald(LAG) | 21.32** | 0.0016 |
| Wald(ERR) | 21.81** | 0.0013 |
| LR(LAG) | 21.23** | 0.0017 |
| LR(ERR) | 21.70** | 0.0014 |
| LR-test for spatial fixed effects | 59.25*** | 0.000 |
| LR-test for time-fixed effects | 1932.21*** | 0.000 |

Note: Robust standard errors in parentheses

*** $p<0.01$

** $p<0.05$

* $p<0.1$.

**Table 9. Regression results of SDM.**

| Variable | ln IGTFP |
|---|---|
| ln DIG | 0.687*** |
|  | (4.04) |
| W*ln DIG | 0.354** |
|  | (2.33) |
| Direct effect | 0.692*** |
|  | (3.97) |
| Indirect effect | 0.319** |
|  | (2.24) |
| Total effect | 1.011** |
|  | (2.21) |
| Control variables | YES |
| City FE | YES |
| Year FE | YES |
| Observations | 2502 |
| R-squared | 0.1014 |
| Log-pseudolikelihood | 2753.8211 |

Note: Robust standard errors in parentheses

*** p<0.01

** p<0.05

* p<0.1.

Therefore, the Durbin spatial model based on spatio-temporal fixed effects is used for analysis in this paper.

**5.3.3 Regression results of the spatial Durbin model.** Table 9 reports the results of the spatial effect estimation. The digital economy coefficient and the spatial interaction term in the SDM model are both significantly positive, and the coefficient of the local digital economy level on the industrial green development of the peripheral areas is 0.354, which is significantly positive, indicating that the local digital economy level can also promote the industrial green development of the peripheral cities. The total effect is further decomposed into direct and indirect effects to explain the spatial spillover effect more accurately. The direct, indirect and total effects of digital economy on industrial green development are 0.692, 0.319 and 1.011, respectively, and all of them are significant. It indicates that the digital economy can not only promote the level of local industrial green development, but also has the spatial diffusion effect of the digital economy level enhancement, which also has a significant role in promoting the industrial green development of peripheral cities. The main reason for this result may be, on the one hand, that the information asymmetry in the market makes the interests of the two parties to market transactions unbalanced, and thus the ability of the market to allocate resources effectively is low [40]. The development of the digital economy reduces the information asymmetry by promoting the dissemination of information, thus increasing the ability of the market to allocate resources effectively and promoting the development of high-quality industry. On the other hand, the improvement of the digital economy level will have an emulation effect on peripheral regions, prompting the governments of peripheral regions to promote the development of the Internet, green innovation and thus the green development of their industries through increased financial investment and other strategies. Hypothesis H3 is further verified.

**Table 10. Robustness check.**

| Variables | ln IGTFP | ln IGTFP | ln IGTFP | ln IGTFP |
|---|---|---|---|---|
| | (1) | (2) | (3) | (4) |
| ln DIG | 0.634** | 0.730** | 1.416*** | 0.787*** |
| | (2.27) | (2.13) | (3.75) | (2.43) |
| Constant | -1.519*** | -0.381 | -1.699*** | -1.497*** |
| | (-3.63) | (-0.56) | (-4.04) | (-2.71) |
| Control variables | YES | YES | YES | YES |
| city FE | YES | YES | YES | YES |
| Year FE | YES | YES | YES | YES |
| Observations | 1946 | 882 | 2502 | 2502 |
| R-squared | 0.081 | 0.104 | 0.110 | 0.646 |
| F-statistic | 8.585 | 5.654 | 12.250 | 13.309 |

Note: Robust standard errors in parentheses

*** p<0.01

** p<0.05

* p<0.1.

## 6. Further discussions

### 6.1 Robustness check

In order to further increase the reliability of our findings, a number of robustness checks were carried out. First, the sample study period is replaced. This study takes 2013–2019 as the sample, 2013 is the year when China's mobile Internet officially entered the 4G era, based on which the digital economy and industrial green development were re-estimated with 2013 as the node, and the impact of digital economy on industrial green development is significantly positive. The regression results are shown in column (1) of Table 10. Second, the regression is re-run excluding cities in the central and western regions. The results show that the digital economy has a significantly positive impact on industrial green development. The regression results are shown in column (2) of Table 10. Third, replace the core explanatory variables. The variable share of computer services and software employees was selected as a new proxy variable for the digital economy, and the regression results are shown in column (3) of Table 10. At this point, the coefficient of the explanatory variable ln DIG is 1.416 and passes the 1% significance level, indicating that the digital economy significantly promotes the green development of industry, proving the reliability of the benchmark regression results.

### 6.2 Endogenous discussion

The main causes of endogeneity problems are mutual causation and omitted variables. According to Huang et al. [75], postal and telecommunication data of Chinese cities in 1984 is selected as an instrumental variable for the digital economy. The impact of infrastructures such as post and telecommunications is inextricably linked to the digital economy, fulfilling the prerequisite of instrumental variable correlation, and the indicator cannot directly affect the greening of industry, fulfilling the requirement of homogeneity of instrumental variables. Since the post office data for each city is cross-sectional and difficult to use directly as an instrumental variable for panel data, the interaction term (HDIG) is constructed as an instrumental variable using the number of post offices per million people in 1984 and the lagged period, i.e., national software revenue from 2010–2018, drawing on Nunn and Qian [80]. Estimated using two-stage least squares (2SLS), the regression results are shown in column (4) in

Table 10. The F-value is 13.309, which is greater than 10 and there is no weak instrumental variable problem. The coefficient of the explanatory variable ln DIG is 0.787 and passes the significance level of 1%, which is consistent with the conclusion of the benchmark regression, and the results of the study further corroborate that the digital economy has a significant contributing effect on industrial green development.

### 6.3 Heterogeneity

To further explore the differential impact of regional digitization levels on the green development of the industrial sector in different regions, Beijing-Tianjin-Hebei, the Yangtze River Delta, the Pearl River Delta, the middle reaches of the Yangtze River, the Central Plains, the Chengdu-Chongqing Urban Agglomeration, municipalities directly under the central government, provincial capitals, and sub-provincial cities as the central city, and the rest of the cities as the peripheral cities, and the basic settings of the variables and the model are kept the same as above. The results of the heterogeneity test are shown in columns (4) and (5) in Table 6. The results show that the digital economy has a significant positive impact on the central city, while the impact on the peripheral cities is not significant. With relatively well-developed infrastructures in central cities, the digital economy can play a full role in promoting the green development of urban industries. On the contrary, there is a digital gap between peripheral cities and central cities, and the green effect of the digital economy is not obvious.

## 7. Conclusions and policy recommendations

### 7.1 Conclusions

In this paper, data related to industrial green development from 2011 to 2019 were collected from 278 cities in China, and the SBM-GML index based on non-expected output and principal component analysis were applied to measure industrial green total factor productivity (IGTFP) and digital economy, respectively. On the basis of theoretical analyses, the relevant hypotheses were verified using the mediating mechanism model and the spatial Durbin model (SDM). The main conclusions are as follows: first, the digital economy can significantly promote the improvement of IGTFP, and the development of the digital economy in the region can drive the green development of industries in the peripheral regions through spatial spillover effects. Second, green technology innovation has a partial mediating role in the process of digital economy affecting IGTFP. Third, industries in central cities can achieve greater green development from digital economy development relative to industries in surrounding cities.

### 7.2 Policy recommendations

First, accelerating the development of digital economy is an important way to promote IGTFP. On the one hand, it is recommended that the government increase investment in digital infrastructure such as 5G mobile networks, strengthen inter-city co-operation, and enhance the positive spatial spillover effect of the digital economy; on the other hand, an industrial internet platform can be set up to promote information sharing and synergy among enterprises, and improve the overall green development of the industry. Secondly, the development of green finance will open up financing channels for enterprises and promote green innovation, thereby reducing energy consumption and environmental pollution and achieving green development. Thirdly, training and education in digital technology for residents and enterprises in peripheral cities. Research institutions, universities and enterprises in central cities are encouraged to cooperate with peripheral cities to promote the application and innovation of digital technologies in peripheral cities.

### 7.3 Suggestions for future studies

This study also has some limitations. First, due to the slow updating of the latest data in some cities, we only used data from 2011–2019 for our analyses, which also led to a failure to fully consider the impact of the COVID-19 epidemic on IGTFP. Future studies should further explore other factors affecting IGTFP using the latest city data. Secondly, the context of China alone may lead to low generalizability of the results of this study. Future studies should further explore other countries.

## Supporting information

**S1 Table. Results of bootstrapping mediation regression analysis.**
(PDF)

**S1 Data.**
(XLSX)

## Author Contributions

**Conceptualization:** Zhengming Wang.

**Data curation:** Xiujin Guo.

**Formal analysis:** Xiujin Guo.

**Funding acquisition:** Xiujin Guo.

**Investigation:** Xiujin Guo.

**Methodology:** Xiujin Guo.

**Project administration:** Zhengming Wang.

**Resources:** Xiujin Guo.

**Software:** Xiujin Guo.

**Supervision:** Zhengming Wang.

**Validation:** Zhengming Wang.

**Visualization:** Zhengming Wang.

**Writing – original draft:** Xiujin Guo.

**Writing – review & editing:** Xiujin Guo.

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
