## [Decision Letter · Decision Letter 0]

1 Jul 2024

PONE-D-24-11461How does the digital economy affect the green development of China's industry?PLOS ONE

Dear Dr. Guo,

Thank you for submitting your manuscript to PLOS ONE. After careful consideration, we feel that it has merit but does not fully meet PLOS ONE’s publication criteria as it currently stands. Therefore, we invite you to submit a revised version of the manuscript that addresses the points raised during the review process.

We look forward to receiving your revised manuscript.

Kind regards,

Zulqarnain Mushtaq, PhD

Academic Editor

PLOS ONE

Journal Requirements:

- https://doi.org/10.1016/j.scitotenv.2022.159428

- https://doi.org/10.1016/j.jbusres.2022.03.041

In your revision ensure you cite all your sources (including your own works), and quote or rephrase any duplicated text outside the methods section. Further consideration is dependent on these concerns being addressed.

"This research was funded by the Jiangsu College Philosophy and Social Science research project under grant number 2023SJYB2214."

"The authors declare no conflicts of interest."

5. In the online submission form, you indicated that [The datasets generated during the current study are available from the corresponding author on reasonable request.]. 

6. We note that [Figures 1 and 2]in your submission contain [map/satellite] images which may be copyrighted. All PLOS content is published under the Creative Commons Attribution License (CC BY 4.0), which means that the manuscript, images, and Supporting Information files will be freely available online, and any third party is permitted to access, download, copy, distribute, and use these materials in any way, even commercially, with proper attribution. For these reasons, we cannot publish previously copyrighted maps or satellite images created using proprietary data, such as Google software (Google Maps, Street View, and Earth). For more information, see our copyright guidelines: http://journals.plos.org/plosone/s/licenses-and-copyright.

a. You may seek permission from the original copyright holder of Figures 1 and 2 to publish the content specifically under the CC BY 4.0 license.  

Reviewers' comments:

Reviewer's Responses to Questions

**Comments to the Author**

1. Is the manuscript technically sound, and do the data support the conclusions?

Reviewer #1: Yes

Reviewer #2: Yes

2. Has the statistical analysis been performed appropriately and rigorously? 

Reviewer #1: Yes

Reviewer #2: Yes

3. Have the authors made all data underlying the findings in their manuscript fully available?

Reviewer #1: Yes

Reviewer #2: Yes

4. Is the manuscript presented in an intelligible fashion and written in standard English?

Reviewer #1: Yes

Reviewer #2: Yes

5. Review Comments to the Author

Reviewer #1: This paper examines the impact of the digital economy on the green development of China's industry. This paper is interesting but not well written. I commend a major revision for under consideration. Some suggestions are as follows:

1. The introduction is not well designed. I suggest that this paper gives why this issue is being studied, how it is being studied, the results of the study, etc., it is necessary to state the main contributions, and how it differs from the rest of the literature.

2. The abstract does not well written and should rewritten it according to this journal.

3. The authors have given some contributions, but these three contributions have been studied in the previous literature. Especially, I suggest authors conduct a compare with the existing papers.

4. The theoretical analysis is confusing and lacks logic. The authors do not conduct an in-depth analysis of its influence mechanism around the topic of this paper.

5. I suggest that the authors should not simply list the literature, but summarize and refine it. In addition, it is recommended to cite some of the latest literature. These papers maybe useful for your revision: Does digital inclusive finance affect urban carbon emission intensity: Evidence from 285 cities in China. Cities. Seeing green: how does digital infrastructure affect carbon emission intensity? Energy Economics.

6. The empirical analysis does not provide an in-depth discussion of its economic implications, that is, the results are not discussed from an economic perspective, including a comparative analysis with the existing literature.

7. Authors need to check the full text.

Reviewer #2: Review report - PONE-D-24-11461

How does the digital economy affect the green development of China's industry?

Major Points

1. The introduction emphasizes the shortcomings of existing research in exploring the impact of the digital economy on industrial green development, and points out that the existing literature lacks systematic research on how the digital economy promotes industrial green development, especially research on its internal mechanisms and spatial spillover effects.

2. Although the importance of the digital economy is mentioned in the introduction, there is a lack of detailed elaboration on how the digital economy specifically affects green development, and the logical relationship between the two is not fully demonstrated.

3.At the end of the introduction, the core issues and innovations of this study are clearly stated, and the unique contribution and practical significance of this study in the existing literature are highlighted.

4.The current summary section of the review lacks a systematic discussion of the shortcomings of existing research. It is recommended to further analyze where existing research is deficient and how these deficiencies affect the conduct of this study.

5. The argumentation chain is incomplete. There is a lack of sufficient argumentation and explanation for the citation of some research conclusions. For example, when mentioning "the important role of digital economy in reducing carbon emissions", you can add some specific mechanism explanations, such as resource optimization allocation, industrial structure upgrading, etc.

6. Inadequate independence of the assumed parts:

Each hypothesis should be formulated more independently and explicitly. The current hypothesis is partially mixed in the literature review, making it easy for readers to confuse the hypothesis with the content of the literature review. It is recommended that hypotheses be listed independently and provide concise theoretical support before each hypothesis.

7.The article uses Moran's I index to test the spatial correlation between the digital economy and industrial greening. It is recommended to provide more specific calculation formulas and steps, as well as an in-depth explanation of the results. Specifically, the construction method of the weight matrix can be shown, and the interpretation of Moran's I value and its application in the spatial Durbin model (SDM) can be analyzed.

8. Although Moran's I test and SDM model selection were conducted, there was no in-depth discussion on the substantive impact mechanism of spatial correlation and its specific application in the model. For example, how does the digital economy affect the green development of neighboring cities through spatial spillover effects? Why is the SDM model more suitable for exploring this spatial effect? It is recommended to consider these to increase your credibility.

9.The article mentions that the model fitting degree (R2 value) increases after adding control variables, indicating that the selection of control variables is reasonable. However, the basis for selecting specific control variables and their mechanism of action in the model are not elaborated in detail. For example, how do control variables relate to the relationship between the digital economy and industrial green development? Please give a specific explanation.

10. Although the intermediary effect of the digital economy in promoting industrial green development through green technology innovation is mentioned, there is a lack of detailed explanation of the definition, measurement method and specific mechanism of GTEC (green technology innovation) in the model. How to ensure that GTEC is a valid mediating variable? Were sensitivity analyzes conducted to verify the robustness of the mediating effects?

6. PLOS authors have the option to publish the peer review history of their article (what does this mean?). If published, this will include your full peer review and any attached files.

Reviewer #1: No

Reviewer #2: **Yes: **Hui Xiao

---

## [Author Response · Author response to Decision Letter 0]

13 Jul 2024

Responses to Reviewer Comments 

Ref: PONE-D-24-11461

Manuscript Title: 

How does the digital economy affect the green development of China’s industry?

 We greatly appreciate the editor and the anonymous reviewers for their insightful comments and constructive suggestions, which immensely helped us improve the manuscript. 

We have revised the paper accordingly. In this document, we respond to questions raised by the reviewers and explain what changes we have made. We hope the review panel finds the revised version satisfactory. All the changes can be seen in the revised manuscript and formatted in red.

In the following, we respond in detail to the comments from the review panel in a point-by-point manner. The key points raised by the reviewers are underlined; our responses and discussions are formatted in blue and bordered.

 For details, please see our submission of the attachment "Response to Reviewers".

Thank you and best regards,

Sincerely

---

## [Decision Letter · Decision Letter 1]

15 Aug 2024

How does the digital economy affect the green development of China's industry?

PONE-D-24-11461R1

Dear Dr. Guo,

We’re pleased to inform you that your manuscript has been judged scientifically suitable for publication and will be formally accepted for publication once it meets all outstanding technical requirements.

Kind regards,

Yihao Li, Doctor

Academic Editor

PLOS ONE

Additional Editor Comments (optional):

Reviewers' comments:

Reviewer's Responses to Questions

**Comments to the Author**

1. If the authors have adequately addressed your comments raised in a previous round of review and you feel that this manuscript is now acceptable for publication, you may indicate that here to bypass the “Comments to the Author” section, enter your conflict of interest statement in the “Confidential to Editor” section, and submit your "Accept" recommendation.

Reviewer #1: All comments have been addressed

2. Is the manuscript technically sound, and do the data support the conclusions?

Reviewer #1: Yes

3. Has the statistical analysis been performed appropriately and rigorously? 

Reviewer #1: Yes

4. Have the authors made all data underlying the findings in their manuscript fully available?

Reviewer #1: Yes

5. Is the manuscript presented in an intelligible fashion and written in standard English?

Reviewer #1: Yes

6. Review Comments to the Author

Reviewer #1: Thank you for the revision. According to the reviewer's opinion, the authors have made a comprehensive revision to the full text, and the revision workload is very heavy. I accept the authors’ changes and consider it acceptable for publication now

7. PLOS authors have the option to publish the peer review history of their article (what does this mean?). If published, this will include your full peer review and any attached files.

Reviewer #1: No

---

## [Editor Report · Acceptance letter]

16 Sep 2024

PONE-D-24-11461R1 

PLOS ONE

Dear Dr. Guo, 

I'm pleased to inform you that your manuscript has been deemed suitable for publication in PLOS ONE. Congratulations! Your manuscript is now being handed over to our production team.

Kind regards, 

on behalf of

Dr. Yihao Li 

Academic Editor

PLOS ONE